# High school science fair: What students say—mastery, performance, and self-determination theory

Frederick Grinnell[1]*, Simon Dalley[2], Joan Reisch[3]

1 Department of Cell Biology, UT Southwestern Medical Center, Dallas, Texas, United States of America,
2 Department of Physics, Southern Methodist University, Dallas, Texas, United States of America,
3 Department of Department of Health Data Sciences and Biostatistics, UT Southwestern O'Donnell School of Public Health, Dallas, Texas, United States of America

* frederick.grinnell@utsouthwestern.edu

## Abstract

Most high school students indicate that participation in science and engineering fairs (SEFs) increased their interest in science and engineering (S&E). The underlying appeal of SEF participation is unknown. However, having this information will help to identify best practices leading to more effective student participation and successful outcomes. To learn more about the appeal of SEF participation, we incorporated into our national SEF surveys a free text *Reason Why?* question asking students the reasons why SEF participation increased or not their interest in S&E. In this paper, we report and analyze the positive and negative comments by 1191 students who participated in our surveys during 2021−22 and 2022−23 and provided free text reasons. The positive reasons that students mentioned most frequently were learned new things; experience doing research; enjoyed/fun experience; and career choice. The negative reasons most frequently mentioned were participation not fun/stressful/boring; not a good project; not interested in science; and required to participate. Overall, students who received coaching and help from scientists made the most positive comments, consistent with our finding that students who received these kinds of help achieved better SEF outcomes. Students who participated in school-only level SEFs made the most negative comments. Reasons students gave why SEF participation increased their interest in S&E aligned with mastery criteria. By contrast, reasons students gave in a previous study regarding why competitive SEFs should be required aligned with performance criteria. Mastery and performance orientations (learning vs. winning) integrate differently with the three elements of self-determination theory: motivation, competence and community engagement. Recognizing these differences in relation to science fair requirements and the S&E career interests of students who participate in SEFs has the potential to enhance the impact of SEF participation on student STEM interest and knowledge.

**Data availability statement:** All relevant data are within the manuscript and its Supporting Information files.

**Funding:** Use of the REDCap survey and data management tool was facilitated by the UTSW Department of Information Resources and Clinical and Translational Science Training Program, NIH grant UL1TR001105. The REDCap funders had no role in study design, data collection and analysis, decision to publish, or preparation of the manuscript.

**Competing interests:** The authors have declared that no competing interests exist.

## Introduction

In recent years, ideas about how best to accomplish science education in the United States have focused increasingly on hands-on practical experience of science and engineering [1–3]. As described in Next Generation Science Standards, student experience of the practices of science is one of three essential dimensions of science education – *students cannot comprehend scientific practices, nor fully appreciate the nature of scientific knowledge itself, without directly experiencing those practices for themselves* [4,5].

Science and engineering fairs (SEFs), which began almost one hundred years ago under the auspices of a civic organization called the American Institute of the City of New York [6], offer one means for students to directly experience the practices of science for themselves. About 5% of U.S. high school students participate in science competitions during high school [7,8]. SEFs have been suggested to potentially promote three important and desirable STEM outcomes: (i) mastery of science and engineering (S&E) practices; (ii) interest in STEM; and (iii) interest in STEM careers [9–11]. Our demographic findings suggested that SEFs can support the goal of STEM education for everyone and not just for the scientists and engineers of the future. Female and male students indicated similar SEF experiences as also reported by others [12–14]. Students from urban and suburban high schools reported similar SEF experiences [15]. And a significant percentage of all students indicated that SEF participation increased their interests in S&E, although outcomes were higher for Asian and Hispanic students (60–70%) compared to Black and White students (40–50%) [15,16].

The idea that SEF participation could have a positive impact on high school students is consistent with research showing that science project-based learning and journal writing advances students' STEM understanding and interests at both the high school [17–22] and undergraduate levels [23–25]. Also, innovative high school programs that combine student participation in SEFs with student and teacher support promote STEM engagement and learning for all students including those from under-represented ethnic minorities and low socioeconomic backgrounds [26–31]. SEFs have become part of science education not only in the United States, but also worldwide [10,11,32–35].

In 2015, we initiated an ethnographic study of high school SEF culture in the U.S. aimed at understanding the experiences of students who participated in SEFs. We approached the research from the perspective of grounded theory [36,37] using quantitative anonymous and voluntary surveys. The overall goal of our research has been to establish a base of knowledge about high school SEFs to help to identify best practices [38]. We learned that about 60% of the students who participate in SEFs are required to do so [15]. When we asked the students whether SEFs should be optional or required, only about 20% indicated SEFs should be required. When asked for reasons why, the most frequently mentioned positive reason to require SEFs was competition incentive (51%). Introduction to the scientific process was rarely mentioned (3%). The most frequently mentioned negative reasons were many students don't like to compete (29.6%) and no enjoyment (17.3%) [39]. Despite

their negative views about a SEF requirement, the majority of students (about 60%) still indicated that SEF participation increased their interest in S&E. The goal of the research described in this paper was to better understand the appeal of SEF participation to these students. Towards this end, we added another question to our surveys asking the students the reason why SEF participation increased or not interest in S&E. In what follows we report and analyze the positive and negative comments of 1191 students who participated in our surveys during 2021−22 and 2022−23 and provided reasons. The positive reasons that students mentioned most frequently were learned new things; experience doing research; enjoyed/fun experience; and career choice. The negative reasons most frequently mentioned were participation was not fun/stressful/boring; not a good project; not interested in science; and required to participate.

Reasons students gave why SEF participation increased student interest in S&E aligned with mastery criteria in contrast to reasons students gave previously regarding whether competitive SEFs should be required, which aligned with performance criteria. Mastery and performance orientations integrate differently with the three elements of self-determination theory: motivation, competence and community engagement. In general, student engagement in STEM research projects fits well with the recent emphasis in STEM education on self-determination theory, which focuses on the importance of student autonomy in decision making, competence to master the work at hand, and connectedness with the STEM community [40,41]. Recognizing these differences in relationship to science fair requirements and the S&E career interests of students who participate in SEFs has the potential to enhance the impact of SEF participation on student's STEM interests and knowledge

## Materials and methods

This study was approved by the UT Southwestern Medical Center IRB (#STU 072014−076). The study design entailed administering to students a voluntary and anonymous online survey using the REDCap survey and data management tool [42]. Survey recipients were U.S. high school students using Scienteer (www.scienteer.com) for online SEF registration, parental consent, and project management during the 2021/22, and 2022/23 school years. Although we treat the Scienteer SEF population as a national group of U.S. high school students, it should be recognized that these students come from seven U.S. states: Alabama, Louisiana, Maine, Missouri, Texas, Vermont, and Virginia. We have no information about the locations where SEFs are held within each of the seven states or state by state student demographics in relation to SEF participation. Because respondents are not be personally identifiable by our surveys, they can share their opinions openly, and we can make the original survey data itself public when we publish the findings and not violate confidentiality.

After giving consent for their students to participate in SEFs, parents can consent for their students to take part in the SEF survey. However, to prevent any misunderstanding by parents or students about a possible impact of agreeing to participate or participating in the survey, access to the surveys is not available to students until after they finish all their SEF competitions. When they initially register for SEFs, students whose parents gave permission are told to log back in to Scienteer after completing the final SEF competition in which they participated. Those who do so are presented with a hyperlink to the SEF survey. Scienteer does not send out reminder emails, and no incentives are offered for remembering to sign back in and participate in the survey. Since 2016, when we began surveying the national Scienteer cohort of SEF students, more than 4,000 students have completed surveys, an overall response rate of about 3%. Given that student participation in the surveys involves an indirect, single electronic invitation without incentive or follow-up, this level of response would be expected [43–45].

The survey used for the current study can be found in supporting information (S1 Survey). The current version is similar to the original survey first adopted in 2015 [46]. However, since then new questions have been added about level of SEF competition; interest in a career in S&E, whether SEF experience increased S&E interest [39]; student ethnicity [16]; and about location of the student's high school (urban, suburban, rural) [15]. Beginning in 2021, following quantitative survey question *20 -- Did your science fair experience increase your interest in the sciences or engineering?* -- we added the free text answer option *20A -- Reason Why?*

The students' answers to the free text answer Reason Why? question is the focus of the current study. Qualitative text analysis was carried out as previously [39,47] using a grounded theory-based approach [37] modeled by NVivo [48,49]. Of the students who completed SEF surveys, approximately 70% (1191) wrote comments about SEF participation and their interests in S&E. Two members of the research team (FG and SD) independently coded students' comments, which were categorized into a matrix of shared student reason groups. The independently coded comments were subsequently revised and harmonized. Longer student comments often expressed more than one reason why, in which case a single student comment was coded into more than one reason group and explains why the total number of grouped comments exceeds the total number of student comments. The complete set of student answers to the *Reason Why?* question and corresponding positive and negative group category assignments can be found in supporting information (S1 Dataset).

Quantitative survey data for the overall student groups and key question categories were determined by frequency counts and percentages. Significance of potential relationships between data items was assessed using Chi-square contingency tables for independent groups. A probability value of p = 0.05 or smaller was accepted as statistically significant but actual p values are shown. No adjustments are made for multiple comparisons.

## Results

### Overview of qualitative survey responses

Beginning in 2021, after the survey question -- *Did your science fair experience increase your interest in the sciences or engineering?* – we added the free text answer option, *Reason Why?* In response, 1191 out of 1790 students provided text comments. S1 Table in supporting information shows that for these 1191 students, the 2021−22 and 2022−23 results were similar regarding demographics, opinions about SEFs, help received, obstacles encountered, and ways of overcoming obstacles. For subsequent analyses of the positive and negative comments by the students, the two years of data were combined.

Tables 1 and 2 and Fig 1 show the frequency and distribution of answers to the *Reason Why?* question organized into positive and negative groups. Overall, positive reasons (1315) outnumbered negative reasons (821). S1 Fig in supporting information shows that the distribution of grouped students' comments was similar from year to year, i.e., 21–22 vs. 22–23.

The top five groups of positive comments accounted for about 83% of all the students' positive comments regarding why SEF participation increased their interest in S&E. The most frequent group fit into the category of non-specific general affirmation, **affirmed – expanded interest** – 308 out of 1315 total. The next most frequent group of positive student comments fit into the category **learned new things** (296 comments), which could mean not only the students' project itself – *seeing your results is always rewarding,* but also, learning about new topics, categories and fields to study. Next came the **experience doing research** (224 comments) such as learning how to set up an experiment or to collect data. One student commented, *made me feel as if I were an actual researcher.* The fourth group with more than 100 comments was **enjoyed/fun experience** (160 comments) ranging from general comments, e.g., *enjoyed learning and doing all of these skills*, to one student's *thrill of getting a project to work.* Finally, the last group was **career choice** (113 comments), which often reflected confirming a student's career interests but sometimes concerned establishing a new trajectory, e.g., *changed my interest in space and environmental studies.* Students also made positive comments but less frequently that fit into the categories **see other proje**cts, **solve society problems**, and **experience presenting research**.

The top five groups of negative comments accounted for about 81% of all the students' negative comments regarding why SEF participation did not increase their interest in S&E. The most frequent examples fit into the category **not fun/ stressful/boring** (179 comments) -- *the experimental process is not something I enjoy*. Next most frequent was **not a good project** (147 comments) for diverse reasons ranging from SEF restrictions -- *the bounds with which the science fair experiment had to be conducted were extremely limiting and did not allow me to pursue my interests* -- to failure of student

**Table 1. Summary of student demographics and experiences on the effect of SEF participation on student interest in S&E.**

| Survey | | | SEF participation increased my interest in S&E | | | Chi Square (p Value) |
|---|---|---|---|---|---|---|
| Question | Answer | # Students | # Yes | # No | % Yes | |
| All Students | | 1191 | 668 | 523 | 56.1 | |
| Grade | 9 | 509 | 295 | 214 | 58.0 | <.001 |
| | 10 | 369 | 165 | 204 | 44.7 | |
| | 11 | 224 | 142 | 82 | 63.4 | |
| | 12 | 88 | 65 | 23 | 73.9 | |
| Ethnicity | Asian | 355 | 238 | 117 | 67.0 | <.001 |
| | Black | 101 | 49 | 52 | 48.5 | |
| | Hispanic | 222 | 141 | 81 | 63.5 | |
| | White | 451 | 205 | 246 | 45.5 | |
| Gender | Female | 641 | 356 | 285 | 55.5 | .749 |
| | Male | 530 | 303 | 227 | 57.2 | |
| Interest in career in S&E | Yes | 712 | 498 | 214 | 69.9 | <.001 |
| | Not sure | 300 | 139 | 161 | 46.3 | |
| | No | 176 | 30 | 146 | 17.0 | |
| SEF requirement | Required | 743 | 352 | 391 | 47.4 | <.001 |
| | Optional | 224 | 190 | 34 | 84.8 | |
| | Project Required | 188 | 114 | 74 | 60.6 | |
| Level of Competition | Beyond school | 710 | 578 | 132 | 81.4 | <.001 |
| | School only | 551 | 452 | 394 | 53.4 | |
| Type of help | Received coaching for interview | 112 | 96 | 16 | 85.7 | <.001 |
| Who helped | Scientists | 82 | 65 | 17 | 79.3 | <.001 |
| | Teachers | 653 | 398 | 255 | 60.9 | .043 |
| | Parents | 622 | 348 | 274 | 55.9 | .955 |

engagement -- *I chose a quite boring project idea due to lack of imagination, creativity, and curiosity*. Next came non-specific **no change in interest** (133 comments). The fourth group **not interested in science** (116 comments) concerned the student's lack of interest in science and engineering, sometimes mentioning non-S&E career trajectories. Finally, dislike of being **required** (89 comments) was the last of the most frequent negative comment groups, especially being required to participate even if not interested in S&E. For instance, *there is no need to force students to do an impractical experiment they don't want to do. It adds a lot of stress to their already busy schedule and lives.*

Students also mentioned limited time and resources in their negative comments but not as frequently as other concerns. Based on the students' quantitative survey answers (S1 Table), about 1/3 participated in SEFs as part of a research team. However, very few students made comments about their project partners. Of those who did, the comments were equally divided between the positive and negative impacts.

Some students wrote just a short phrase in answer to the *Reason Why?* question. However, many students expressed their thoughts in complex sentences. When a student comment expressed more than one reason why, then the comment was coded into more than one comment group, which is why the total number of grouped comments (2136) almost doubled the total number of students' comments (1191). Also, occasionally a student's comments could be put into more than one group. Tables 3 and 4 show examples of student comments for each of the groups listed in Table 2 illustrating the complexity. Italics shows the text based on which the comments were assigned to a particular group. For instance, in the first entry in Table 3, the phrases *deeper understanding, topic I enjoy, and want to keep on studying* resulted respectively

**Table 2. Frequency of free text student comments in answer to the *Reason Why?* question.**

| Overall | Grouped Comments | # Comments | % of Overall Group | Group ID |
|---|---|---|---|---|
| Positive (1315 reasons) | Affirmed/expanded interest | 308 | 23.4 | P1 |
| | Learned new things | 296 | 22.5 | P2 |
| | Experience doing research | 224 | 17.0 | P3 |
| | Enjoyed/fun experience | 160 | 12.2 | P4 |
| | Career choice | 113 | 8.6 | P5 |
| | See other projects | 77 | 5.9 | P6 |
| | Solve society problems | 76 | 5.8 | P7 |
| | Experience presenting research | 54 | 4.1 | P8 |
| | Positive experience with project partners | 7 | 0.5 | P9 |
| Negative (821 reasons) | Not fun/stressful/boring | 179 | 21.8 | N1 |
| | Not a good project | 147 | 17.9 | N2 |
| | No change in interest | 133 | 16.2 | N3 |
| | Not interested in science | 116 | 14.1 | N4 |
| | Required | 89 | 10.8 | N5 |
| | Not enough time | 39 | 4.8 | N6 |
| | Too much work | 31 | 3.8 | N7 |
| | Negative judging and teachers | 24 | 2.9 | N8 |
| | Waste of time | 17 | 2.1 | N9 |
| | Inadequate resources | 16 | 2.0 | N10 |
| | Didn't learn anything | 16 | 2.0 | N11 |
| | Not good at science | 8 | 1.0 | N12 |
| | Negative experience with project partners | 6 | 0.7 | N13 |

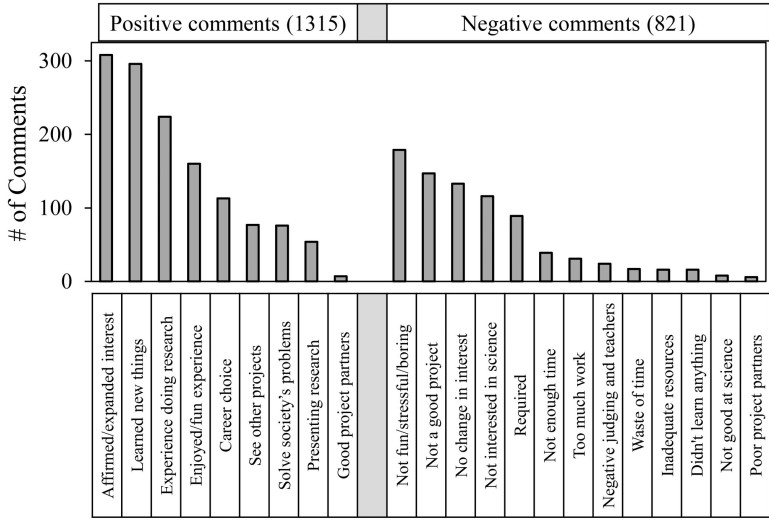

**Fig 1. Distribution of students' comments in answer to the *Reason Why?* question.**

**Table 3. Examples and group assignments of positive student comments in answer to the *Reason Why?*.**

| Examples of Positive Comments | Group ID |
|---|---|
| It gave me a *deeper understanding on a topic I enjoy*, and it made me *want to keep on studying about it*. | P2, P4, P1 |
| It helped me *develop further interest* after being able to research and *actually experience how to research and collect data* in a lab | P1, P3 |
| I have *always wanted to major* and *pursue a career* in science. | P1, P5 |
| Saw *tons of cool projects* there which made me *gain interest*. | P6, P1 |
| It is always inspiring to *see what others do* with science fair and it gives a *hands on experience*. | P6, P3 |
| Science fair allowed me to realize how science can be applied to *make the world a better place*. | P7 |
| It allowed me to *go deeper into a concept* and overall. I *really enjoy* working on a project for a year then *presenting that project*. | P2, P4, P8 |
| Although making the project was sometimes stressful, it was *still fun*. I got to *research new things* and *meet a good partner*. | P4, P3, P9 |

**Table 4. Examples and group assignments of negative student comments in answer to the *Reason Why?*.**

| Examples of Negative Comments | Group ID |
|---|---|
| The Science fair *didn't seem fun* it just seemed like an assignment I was *forced to do*. I do not look forward to doing a science fair in the future. I was already interested in biology but the biology *topics I wanted* to were *shut down by my teacher*. | N1, N5, N2, N8 |
| No, because the *project did not spark* my area of interest in science. | N2, N3 |
| I'm *more interested and dedicated* to politics and business rather than the STEM and sciences. | N4 |
| Honestly, it felt like *unnecessary work added* onto my already large pile of work for this and other classes. It felt *stressful* and I *didn't really get anything* out of it. | N5, N1, N11 |
| The *time pressure* and *stress* sometimes became overwhelming. | N6, N1 |
| Because I like science but *I don't like the amount of work*. | N7 |
| Free *time wasted+ not helpful* | N9, N11 |
| With *time and material constraints*, I wasn't able *to pick a topic I enjoyed*. | N6, N10, N2, N1 |
| I'm *not good at it* and science *isn't something I'd like to do* in the future | N12, N4 |
| My partner and I *did not get along* and the results of the project were blamed on me, so I *did not enjoy* doing the science fair. | N13, N1 |

to the comment assignments **P2 (learned new things)**, **P4 (enjoyed/fun experience)** and **P1 (affirmed – expanded interest)**. In the first entry in Table 4 (negative student comment groups), the phrases *didn't seem fun, forced to do, topics I wanted*, and *shut down by my teacher* resulted respectively to the comment assignments **N1 (not fun/stressful/boring)**, **N5 (required)**, **N2 (not a good project)** and **N8 (negative judging and teachers)**.

### Qualitative survey responses in relationship to key question categories -- students' interests and experiences

Table 1 shows demographic information about SEF outcomes and experiences of the group of students who answered the *Reason why?* question. Consistent with previously reported findings [15,16,39], overall more than half the students indicated that SEF participation increased their interests in S&E (56.1%) which was similar to the percent of positive reasons given by students (61.6%, Table 2). Students in older grades were more positive compared to younger students. Asian and Hispanic students were more positive than Black and White students. No gender differences were observed. Positive SEF experience outcomes correlated with student interest in S&E careers; optional vs. required SEF participation; SEF competition beyond the school-only level; coaching for the interview; and help from scientists.

Subsequent tables and figures report the frequency and distribution of students' positive and negative comments in relationship to student demographics and SEF experiences. The tables show the numbers of students in each experience group and their overall positive and negative comments. The bar graphs beneath the tables show the distribution data for the most frequently selected *Reason Why?* categories as comments per 100 students in the group. S2 Table in supporting information shows the number of comments per 100 students numerically.

Fig 2 shows that when the students' comments were sorted according to whether they answered quantitatively *yes* or *no* to the question of whether SEF participation increased their interests in S&E, then 98% of the comments that sorted into the positive groups were from students who answered *yes* compared to 13% positive of the students who answered *no*. Distribution data (# comments/100 students) showed that in every category, the frequency of qualitative comments was similar to the students' overall quantitative responses.

Fig 3 shows that the overall frequency of qualitative comments also corresponded, albeit not as closely, with whether students indicated an interest in an S&E career. Of the 712 students who indicated they were interested in an S&E career, almost 73% made positive comments about SEF participation but 27% made negative comments. Conversely, of the 176 students who indicated no interest in an S&E career, about 78% made negative comments but 22% were positive. Those unsure about careers were in between -- 55% positive comments and 45% negative.

Distribution data showed that students who were uninterested in an S&E career made the fewest positive comments in every category. They made the most negative comments about **not interested in science** and also were more likely to comment negatively that SEF participation was **not fun/stressful/boring** and about SEF participation being **required.**

Fig 4 shows the frequency and distribution of students' comments in relation to whether students were required to participate in SEFs. The more autonomous and self-motivated students, as measured by whether they chose to participate in SEFs, the more likely they made positive comments about their SEF experiences. For the 224 students for whom participation in SEFs was optional, the frequency of positive comments was 89.3% compared to 52.6% of the 743 students required to participate in SEFs. For the 188 students who were required to carry out a school project and chose SEF participation as that project, 65.1% made positive comments.

| Survey Question | Survey Answer | # (%) Students | Comments | |
|---|---|---|---|---|
| | | | Positive # (%) | Negative # (%) |
| All Students | | 1191 (100) | 1315 (61.6) | 821 (38.4) |
| SEF participation increased my interest in S&E | Yes | 668 (56.1) | 1192 (98.3) | 21 (1.7) |
| | No | 523 (43.9) | 123 (13.3) | 800 (86.7) |

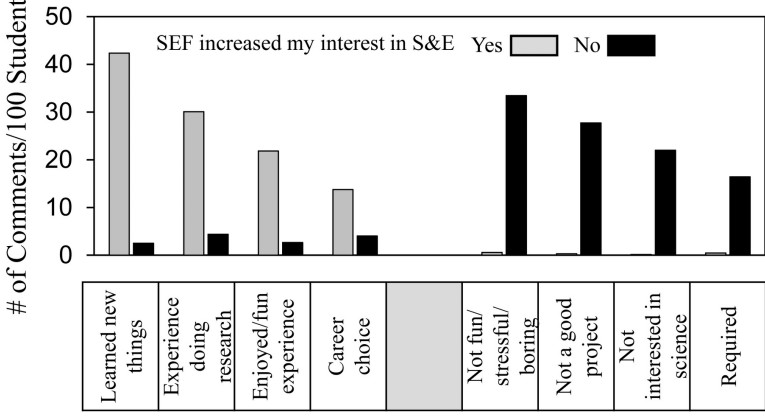

**Fig 2. Frequency and distribution of free text student comments sorted according to student's *yes* or *no* answers to the SEF participation outcome question.**

| Survey Question | Survey Answer | # (%) Students | Comments | | | |
|---|---|---|---|---|---|---|
| | | | Positive # (%) | | Negative # (%) | |
| Interest in a career in S&E | Yes | 712 (59.9) | 979 | (73.1) | 360 | (26.9) |
| | No | 176 (14.8) | 66 | (22.4) | 228 | (77.6) |
| | Unsure | 300 (25.3) | 267 | (54.5) | 223 | (45.5) |

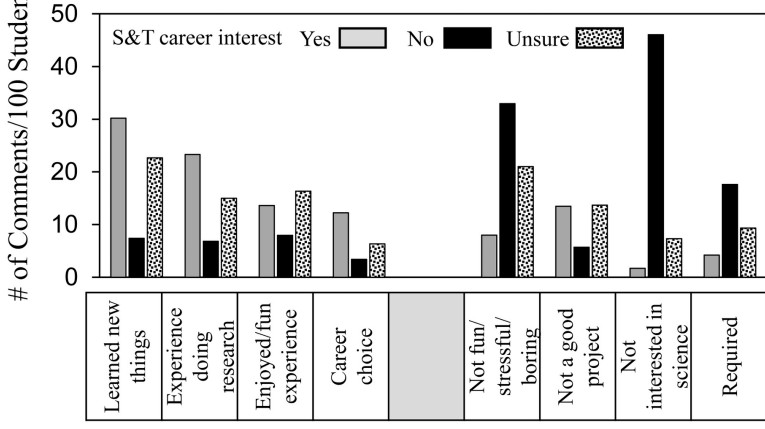

**Fig 3. Frequency and distribution of students' comments depending on students' interests in a S&E career.**

The overall distribution pattern of positive comments was similar compared to all students (Fig 1) except students for whom SEFs were optional commented more regarding **learned new things** and the **experience doing research**. Students required to carry out a school project who chose SEF participation made as many positive comments about enjoying SEF participation and career choice as did those students for whom SEF participation was completely optional. Unexpectedly, the distribution pattern of negative comments showed a more binary response pattern. Unlike students for whom SEF participation was required or a project was required, students who chose to participate in SEFs made very few negative comments in any reason category regarding SEF participation.

Fig 5 shows the frequency and distribution of students' comments in relation to whether students received help from scientists or coaching for the interview. More than any other types of help, students who receive coaching and help from scientists achieve better quantitative SEF outcomes [16] including indicating that SEF participation increased student interest in S&E [39]. (See Table 1) Both types of help had a positive influence -- 85.7% vs. 59.5% and 88.3% vs. 58.7% respectively -- on the frequency of positive student comments.

The overall distribution pattern of positive comments differed somewhat for help from scientists and coaching. Students who received help from scientists were especially likely to make more positive comments about the **experience doing research** and **career choice**. Students who received coaching also made more positive comments about **learned new things**. Both types of help resulted in a high level of student satisfaction with their SEF experience judging from the low number of negative comments in any category.

Fig 6 shows the frequency and distribution of students' comments in relation to whether students participated in school-only or more advanced SEFs, i.e., district, regional or state levels. Not all students indicated the SEF level in which they

| Survey question | Survey answer | # (%) Students | Comments | | | |
| --- | --- | --- | --- | --- | --- | --- |
| | | | Positive # (%) | | Negative # (%) | |
| Science fair requirement | Optional | 224 (19.0) | 369 | (89.3) | 44 | (10.7) |
| | Required | 743 (62.9) | 691 | (52.6) | 623 | (47.4) |
| | Project | 188 (15.9) | 229 | (65.1) | 123 | (34.9) |

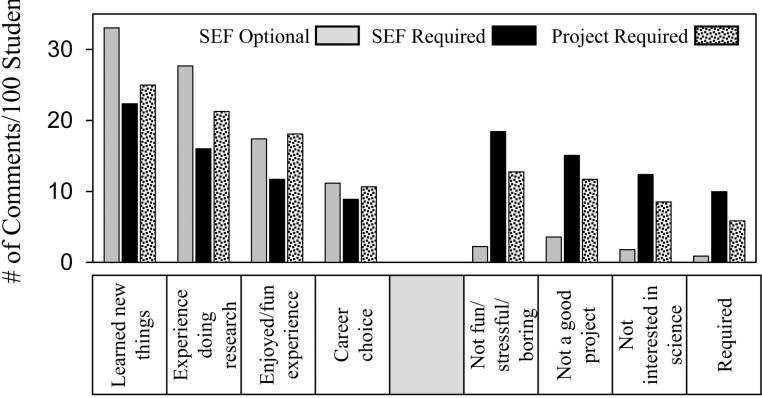

**Fig 4. Frequency and distribution of students' comments depending on SEF participation requirement.**

participated, which is why the number of students is lower than in the other comparisons. About half the students who commented participated in school-only SEFs. The other half advanced to district, region and state SEF levels.

Students who participated in more advanced SEFs made twice as many positive comments about SEF participation compared to those who participated at the school only level -- 84.5% vs 44.7%. Indeed, except for the group of students who indicated no interest in S&E careers (see Fig 3), the students who participated in school-only SEFs were the only ones in the current study for whom less than half the student comments were positive about their experience. The distribution data showed students who competed in more advanced SEF competitions were more likely to comment positively and less likely to comment negatively in every category.

Fig 7 shows the frequency and distribution of students' comments in relation to grade level. Previously, we learned from survey data that most students participate in SEFs during 9th and 10th grades, whereas students in 11th and 12th grades were more likely to be interested in S&E careers and to indicate that SEF participation increased their interests in S&E [50] (See Table 1). Students in 11th and 12th grades made more positive comments explaining why SEF participation increased their interest in S&E compared to students from 9th and 10th grades.

According to the distribution results, 12th graders made more positive comments in every category, whereas 11th graders especially mentioned **learned new things** and the **experience doing research** as reasons that SEF participation increased their interest in S&E. Students in 12th grade also were the least likely to make negative comments except some indicated an interest in careers other than in the sciences, which we reconfirmed in the comment dataset (S1 Dataset). Students in 9th and 10 grades were most likely to comment that SEF participation was **not fun/stressful/boring** as the reason why SEF participation did not increase their interest in S&E.

Fig 8 shows the frequency and distribution of students' comments in relation to student ethnicity. Students of all ethnicities were positive overall, but Asian and Hispanic students made more positive comments than Black and White students

| Survey question | Survey answer | # (%) Students | Comments | |
|---|---|---|---|---|
| | | | Positive # (%) | Negative # (%) |
| Help from scientists | Yes | 82 (6.9) | 144 (85.7) | 24 (14.3) |
| | No | 1109 (93.1) | 1171 (59.5) | 797 (40.5) |
| Received coaching | Yes | 112 (9.4) | 182 (88.3) | 24 (11.7) |
| | No | 1080 (90.6) | 1133 (58.7) | 797 (41.3) |

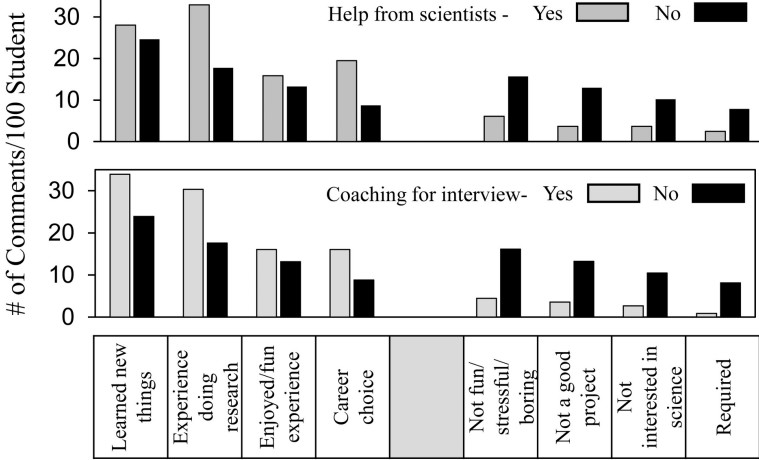

**Fig 5. Frequency and distribution of students' comments depending on coaching and help from scientists and teachers.**

consistent with previous quantitative survey findings [15,16]. According to the distribution results, Asian and Hispanic students were more likely to comment positively about **learned new things** as the reason why SEF participation increased their interests in S&E. Asian students also were more likely than others to comment positively about the **experience doing research**. Black students were the least likely to comment positively about **career choice**. On the negative side, Black and White students were most likely to complain that SEF participation was **not fun/stressful/boring** as the reason why SEF participation did not increase their interest in S&E, and White students were most likely to comment **not a good project**.

## Discussion

In this paper, we present findings based on free text comments by 1191 students who participated in our SEF surveys during 2021−22 and 2022−23. Almost 60% of these students indicated that SEF participation increased their interest in S&E. When asked as a free text question the *Reason Why?* SEF participation increased or not their interest in S&E, students provided 1315 positive reasons and 821 negative ones. Aside from non-specific affirmed/expanded interest (23.4%), the positive reasons most frequently mentioned were learned new things (22.5%); experience doing research (17.0%); enjoyed/fun experience (12.2%); and career choice (8.6%). The most frequently mentioned negative reasons students gave as to why SEF participation did not increase their interest in S&E, aside from non-specific no change in interest (16.2%), were participation was not fun/stressful/boring (21.8%); not a good project (17.9%); not interested in science

| Survey question | Survey answer | # (%) Students | Comments | | | |
|---|---|---|---|---|---|---|
| | | | Positive # (%) | | Negative # (%) | |
| Competition level | Beyond school | 491 (41.1) | 739 | (84.5) | 145 | (16.4) |
| | School only | 551 (46.3) | 432 | (44.7) | 530 | (55.1) |

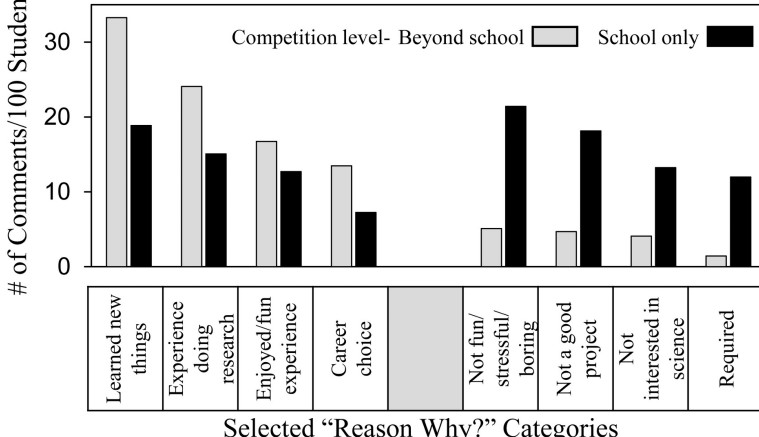

**Fig 6. Frequency and distribution of students' comments depending on SEF competition level.**

| Survey question | Survey answer | # (%) Students | Comments | | | |
|---|---|---|---|---|---|---|
| | | | Positive # (%) | | Negative # (%) | |
| Grade in which students participated in SEFs | 9 | 509 (42.8) | 563 | (62.5) | 338 | (37.5) |
| | 10 | 369 (31.0) | 338 | (51.4) | 319 | (48.6) |
| | 11 | 224 (18.8) | 281 | (67.7) | 134 | (32.3) |
| | 12 | 88 (7.4) | 132 | (81.5) | 30 | (18.5) |

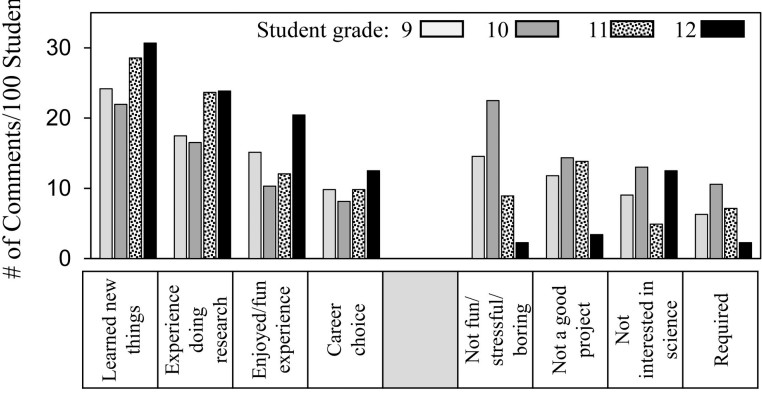

**Fig 7. Frequency and distribution of positive and negative comments depending on student grade.**

| Survey question | Survey answer | # (%) Students | | Comments | | | |
|---|---|---|---|---|---|---|---|
| | | | | Positive # (%) | | Negative # (%) | |
| Student ethnicity | Asian | 354 | (31.4) | 472 | (72.6) | 178 | (27.4) |
| | Black | 101 | (9.0) | 98 | (54.7) | 81 | (45.3) |
| | Hispanic | 222 | (19.7) | 263 | (69.0) | 118 | (31.0) |
| | White | 451 | (40.0) | 418 | (51.6) | 392 | (48.4) |

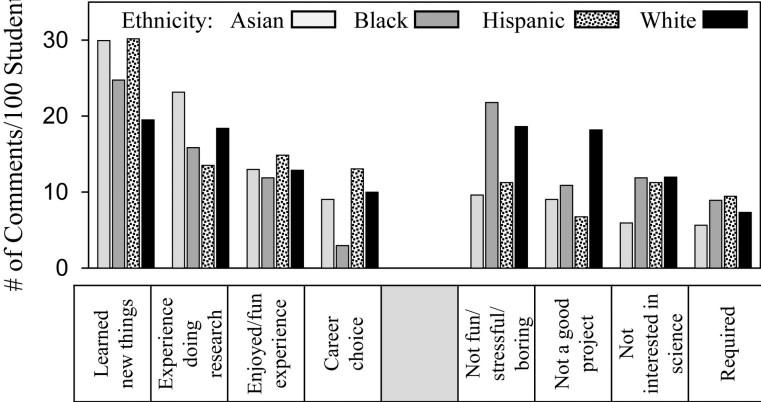

**Fig 8. Frequency and distribution of students' comments depending on ethnicity.**

(14.1%); and required to participate (10.8%). Ironically, although time pressure was mentioned in quantitative surveys as a SEF obstacle by more than 60% of students and to a greater extent than any other obstacle [16] (see Table 1), less than 5% of students mentioned time constraints as their reason why SEF participation did not increase their interest in S&E.

Several limitations of our study are worth noting. We treat the Scienteer SEF population as a national group but these students may not be truly representative of a national sample since they come from only 7 U.S. states and only attend high schools where SEFs are available about which we have no information. Moreover, the 3% response rate of survey respondents may not be truly representative of the high school student population as a whole participating in SEFs. Nevertheless, the answers of the cohort of students surveyed through Scienteer regarding their opinions about whether SEFs should be required, sources of help, types of help, obstacles encountered, and means of overcoming obstacles closely overlap from year to year. In addition, most answers also overlap with regional high school students we surveyed using direct survey distribution with a 57% response rate [46,47] and with undergraduate science students surveyed to reflect back on their high school SEF experiences [50].

The students' positive comments match well the ideal goals of SEF participation [9–11] but differ markedly from the surveys carried out previously in which we asked students if competitive SEF participation should be optional or required [39]. In that case, the students' answers tended to focus on the competition not the SEF learning experience, i.e., the most frequently mentioned positive reason was competition incentive (50%); the most frequently mentioned negative reason was many students don't like to compete (29.6%).

We suggest that these two types of student answers about SEF participation reflect two different ways to understand the educational value of SEF participation. On one hand, mastery orientation, i.e., emphasis on understanding and

improving STEM skills and knowledge; and on the other hand, performance orientation, i.e., competing with others with an emphasis on demonstrating high ability [51–53]. Mastery and performance orientations reflect the two different objectives of the National Research Council Framework definition underlying NGSS -- science for everyone and science for the scientists and engineers of the future [4].

Mastery and performance orientations integrate differently with the three elements of self-determination theory: motivation, competence and community engagement [40,41]. Mastery more clearly aligns with intrinsic motivation and demonstrating competence but not necessarily with community engagement. Performance more clearly aligns with extrinsic motivation, demonstrating competence, and enhanced community engagement. Recognizing these differences in relationship to the interests of high school students who participate in SEFs has the potential to enhance the impact of SEF participation on student STEM interests and knowledge.

Overall, more than half the students (61.6%) made positive comments about why SEF participation increased their interest in S&E but with important nuances. Students interested in a S&E career made more positive comments (74.6%) compared to those uninterested (5%) or unsure (20.4%). Students for whom SEF participation was optional made more positive comments (89.3%) compared to those for whom SEF participation was required (53.6%) or a school project was required (65.1%). Students in older grades made more positive comments compared to younger students (74.6 vs. 57%). Older students were more likely to mention learned new things and the research experience as reasons. For younger students not fun stood out as a negative reason. The differences above add support to key SEF recommendations previously suggested. (i) Promote student autonomy -- incentivize rather than require SEF participation currently required for more than 60% of high school students nationwide [15]. (ii) Promote student competence – since most high school students participate in SEFs in 9th and 10th grades (75%), but the younger students are less interested in S&E, recognize and develop different SEF objectives and assessments for 9th/10th vs. 11th/12th grades [50].

Asian and Hispanic students made more positive comments than Black and White students (78% vs. 53.2%). Especially Black students were less likely than others to write that SEF participation increased their interests in S&E because of career choice. Students who received coaching for the interview or help from scientists made more positive comments (88.3% and 85.7%) than students who did not have these opportunities (58.7 and 59.5%). Career choice was mentioned frequently by these students as the reason SEF participation increased their interest in S&E, which emphasizes the value of connectedness with the STEM community. We suggest that Black students, and indeed all students, would benefit from programs to increase coaching for the SEF interview and help from scientists, now available to less than 10% of students [16] (See Table 1). The success of innovative programs to provide the latter kind of support shows the possibility and importance of developing such programs [26–31].

Not a good project was by far the most important reason that White students said SEF participation did not increase their interest in S&E. In general, not a good project meant limitation in the project the student was allowed to do or assigned. The comments about not a good project are particularly concerning because in a sense not a good project represents SEF project failure, the opposite of competence. From the perspective of self-determination theory, being required to participate in SEFs and in addition being assigned a failing project not of one's own choosing undermines the positive value of motivation more than any other combination.

Based on the frequency and distribution of negative reasons why SEF participation did not increase student interest in S&E, students who participated in school-only SEFs had a SEF experience worse than any other overall group. The number of students who can advance to SEFs beyond the school-only level will depend not only on student performance, but also on the number of SEF entries that a school or district permits to advance. How many typically will be governed by individual school and district policies. If school-only SEFs are the other possible option, then other SEF structures might be more successful such as incorporating SEFs into community STEM festivals [54,55].

In conclusion, based on free text comments by 1191 students who participated in our national science and engineering fair (SEF) surveys during 2021−22 and 2022−23, we have learned new information about SEF participation

and student interest in S&E. Reasons that students gave why SEF participation increased their interest in S&E aligned with mastery criteria. By contrast, reasons students gave in a previous study regarding whether or not competitive SEFs should be required aligned with performance criteria [39]. Mastery and performance orientations integrate differently with the three elements of self-determination theory: motivation, competence and community engagement. Recognizing these differences in relation to science fair requirements and the S&E career interests of students who participate in SEFs has the potential to enhance the impact of SEF participation on student STEM interest and knowledge.

## Supporting information

**S1 Survey.  Survey questions.**
(PDF)

**S1 Table.  Quantitative student responses to the SEF survey year by year.**
(PDF)

**S2 Table.  # comments/100 students in relationship to student demographics and SEF experiences.**
(PDF)

**S1 Fig.  Year by year distribution of students' comments in answer to the *Reason Why?* question.**
(PDF)

**S1 Dataset.  Complete set of student answers to the *Reason Why* question and corresponding positive and negative group category assignments.**
(XLSX)

## Acknowledgments

FG holds the UTSW Robert McLemore Professorship. We are grateful to Russell Cowen and Rocky Slavin who were managers of Scienteer Technologies at the time we carried out our surveys and incorporated the parental consent and SEF survey REDCap links into the Scienteer website.

## Author contributions

**Conceptualization:** Frederick Grinnell, Simon Dalley.

**Data curation:** Frederick Grinnell, Simon Dalley.

**Formal analysis:** Frederick Grinnell, Simon Dalley, Joan Reisch.

**Funding acquisition:** Frederick Grinnell.

**Investigation:** Frederick Grinnell, Simon Dalley.

**Methodology:** Frederick Grinnell, Simon Dalley, Joan Reisch.

**Project administration:** Frederick Grinnell.

**Resources:** Frederick Grinnell.

**Supervision:** Frederick Grinnell.

**Validation:** Frederick Grinnell, Simon Dalley, Joan Reisch.

**Writing – original draft:** Frederick Grinnell.

**Writing – review & editing:** Simon Dalley.

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
