## [Decision Letter · Decision Letter 0]

21 Apr 2025

Dear Dr. Grinnell,

Thank you for submitting your manuscript to PLOS ONE. After careful consideration, we feel that it has merit but does not fully meet PLOS ONE’s publication criteria as it currently stands. Therefore, we invite you to submit a revised version of the manuscript that addresses the points raised during the review process.

 After careful consideration of the reviewers' feedback, I have determined that only minor revisions are required at this stage. Please revise the manuscript accordingly and submit the updated version along with a detailed response to the reviewers' comments. We look forward to receiving your revised manuscript.

We look forward to receiving your revised manuscript.

Kind regards,

Mohammad Ali Yousef Yamin, Ph.D

Academic Editor

PLOS ONE

2. Please update your submission to use the PLOS LaTeX template. The template and more information on our requirements for LaTeX submissions can be found at "http://journals.plos.org/plosone/s/latex"

Additional Editor Comments (if provided):

Reviewers' comments:

Reviewer's Responses to Questions

**Comments to the Author**

1. Is the manuscript technically sound, and do the data support the conclusions?

Reviewer #1: Yes

Reviewer #2: Yes

2. Has the statistical analysis been performed appropriately and rigorously?

Reviewer #1: Yes

Reviewer #2: Yes

3. Have the authors made all data underlying the findings in their manuscript fully available?

Reviewer #1: Yes

Reviewer #2: Yes

4. Is the manuscript presented in an intelligible fashion and written in standard English?

Reviewer #1: Yes

Reviewer #2: Yes

Reviewer #1: The abstract needs revision to enhance clarity, coherence, and alignment with academic standards. Please address the following:

1. The objective of the study is not clearly stated at the beginning. Please explicitly state the research question or purpose of the study in the first 1–2 sentences.

2. Strengthen the final statement by summarizing how your findings contribute to existing knowledge or suggest practical implications for science fair program development.

The introduction presents useful background information but requires revision for better clarity, coherence, and focus. Please consider the following specific suggestions:

1. While the context of SEFs is well-described, the specific research gap and aim of this study are buried in the last paragraph. Please clearly articulate the research objective and contribution earlier in the introduction (ideally in the second or third paragraph).

2. some points (e.g., the benefits of coaching/help from scientists, and participation differences by demographics) are repeated multiple times. Keep these concise and focused for the introduction and elaborate in the results/discussion section.

The Materials and Methods section is generally informative, but the following revisions are necessary to enhance clarity, rigor, and replicability:

1. While a 3% response rate is acknowledged, this low rate may raise concerns about response bias. Please briefly discuss potential limitations this poses for generalizability.

2. The mention of chi-square testing is brief. Please specify (Which variables were tested for association. Whether assumptions for the chi-square test (e.g., minimum expected cell counts) were checked.

The Discussion section provides thoughtful insights and successfully interprets both qualitative and quantitative findings. However, the section would benefit from clearer organization, strengthened coherence, and a deeper discussion of implications. Below are detailed suggestions:

1. Repetition of certain statistics ( number of comments, percentages) can be reduced in the discussion since they were already reported earlier.

2. The limitation regarding geographic coverage (seven states) is mentioned, but its potential effect on the findings ( regional variation in STEM programs, demographics) should be briefly discussed to show awareness of context sensitivity.

3. Discuss the potential for targeted interventions to support underrepresented groups, such as Black students, who reported less benefit from SEF participation.

Reviewer #2: Dear Author/s

Thank you for the opportunity to review your manuscript. I would like to commend your efforts in addressing an important and timely topic related to Science and Engineering Fair (SEF) participation and its impact on students’ interest in STEM fields.

I have provided a set of detailed comments and constructive suggestions, organized by each section of the manuscript. These observations are intended to enhance the clarity, coherence, theoretical grounding, and overall academic rigor of your paper.

Please note that these revisions are not meant to undermine your valuable contributions, but rather to help strengthen the quality of your research and ensure it meets the highest standards of scholarly publication. Addressing these points carefully will not only improve the impact of your study but also help readers better understand the implications of your findings.

Abstract

The abstract should include a clear statement of the study’s purpose or research question.

Underdeveloped Implications, the significance of the findings and their implications for educators, policymakers, or STEM programs are not clearly stated. Please try to refrase it to show your implications

Introduction

- The introduction should clearly define the research gap the current study seeks to address.

- Explicitly stating the research questions or hypotheses would help define the focus of the study.

Materials and Methods

- While the paper acknowledges that the data only come from seven U.S. states, it should provide more detail about student demographics and SEF contexts within those states.

- The survey’s evolution is described, but a summary of the key question categories in the main text (not only the appendix) would aid understanding.

- A 3% response rate is mentioned as typical for this format, but a brief discussion on how non-response bias might affect findings would be valuable.

Results

Ethnic and Grade-Level Differences are mentioned, but clearer comparative analysis (perhaps with tables) would improve readability and interpretation. Optional)

While chi-square results are reported, actual test statistics are not included. Adding these would improve transparency.

Discussion

The potential impact of SEF structures (e.g. school-only vs. district-level fairs) on student outcomes is mentioned but could be developed into concrete recommendations.

Conclusion

Please ensure that the conclusion directly aligns with the objectives stated (or to be stated) in the introduction.

general recommendation

The paper presents important insights and has the potential to contribute meaningfully to the literature on science fairs and student motivation in STEM. However, revisions are necessary to improve theoretical framing, clarity, and analytic depth. I recommend major revisions before the paper is considered for publication

**Do you want your identity to be public for this peer review?** For information about this choice, including consent withdrawal, please see our Privacy Policy

Reviewer #1: **Yes: ** . Mohammad Ibrahim Sweiss

Reviewer #2: **Yes: ** nour taher Alaqra

---

## [Author Response · Author response to Decision Letter 1]

8 May 2025

We appreciated the reviewers’ helpful comments about how to make our paper clearer and have made revisions along the lines that they suggested as summarized below.

Response to Reviewer #1.

1) The Abstract has been revised with the objective of the study clearly stated at the beginning and an explanation of how the findings contribute to existing knowledge at the end.

2) The Introduction has been revised with the research objective and contribution presented earlier and points about benefits of coaching etc. moved to the Results and Discussion sections.

3) Instead of the Materials and Methods, the second paragraph of the Discussion now points out and comments on the various limitations of the study including the 3% response rate and geographic coverage.

4) Chi-square statistics mentioned in Materials and Methods refers to Table 2, which has been moved from Supplemental data to the Results section as requested by Reviewer #2 -- "a summary of the key question categories in the main text (not only the appendix) would aid understanding."

5) Repetition of certain statistics (comments and percentages) has been maintained in the Discussion so that readers can follow the text without turning back to earlier sections.

6) The Discussion section has been revised to present more clearly the potential for targeted interventions to support underrepresented groups such as Black students.

Response to Reviewer #2.

The concerns of Reviewer #2 overlap closely with those of Reviewer #1 and have been addressed as follows:

1) Clearer statement of the study’s research question and significance of the findings in the revised Abstract and Introduction

2) Clearer statement of research gap the study seeks to address in the revised Abstract and Introduction

3) Data coming from only seven U.S. states and student demographics and SEF contexts within those states and 3% response rate. – New paragraph regarding limitations in the Discussion (2nd paragraph).

4) Summary of key question categories -- now Table 2 (chi square results) -- in the Results and not the Supplemental info.

5) The potential impact of SEF structures developed into concrete recommendations in the revised Discussion

---

## [Editor Report · Decision Letter 1]

12 May 2025

High School Science Fair: What Students Say -- Mastery, Performance, and Self-Determination Theory

PONE-D-25-09082R1

Dear Dr. Frederick Grinnell

We’re pleased to inform you that your manuscript has been judged scientifically suitable for publication and will be formally accepted for publication once it meets all outstanding technical requirements.

Kind regards,

Mohammad Ali Yousef Yamin, Ph.D

Academic Editor

PLOS ONE
---

## [Editor Report · Acceptance letter]

PONE-D-25-09082R1

PLOS ONE

Dear Dr. Grinnell,

I'm pleased to inform you that your manuscript has been deemed suitable for publication in PLOS ONE. Congratulations! Your manuscript is now being handed over to our production team.

Kind regards,

on behalf of

Dr. Mohammad Ali Yousef Yamin

Academic Editor

PLOS ONE